# Seasonality of Rotavirus Hospitalizations at Costa Rica’s National Children’s Hospital in 2010–2015

**DOI:** 10.3390/ijerph16132321

**Published:** 2019-06-30

**Authors:** Katarina Ureña-Castro, Silvia Ávila, Mariela Gutierrez, Elena N. Naumova, Rolando Ulloa-Gutierrez, Alfredo Mora-Guevara

**Affiliations:** 1Servicio de Pediatría, Hospital William Allen Taylor, Caja Costarricense del Seguro Social (CCSS), Turrialba 30501, Costa Rica; 2Posgrado de Pediatría, Universidad de Costa Rica (UCR) & Caja Costarricense de Seguro Social (CCSS), San José 2060, Costa Rica; 3Servicio de Emergencias, Hospital Nacional de Niños “Dr. Carlos Sáenz Herrera”, Centro de Ciencias Médicas, Caja Costarricense de Seguro Social (CCSS), San José 10103, Costa Rica; 4Division of Nutrition Data Science, Friedman School of Nutrition Science and Policy, Tufts University, Boston, MA 02111, USA; 5Servicio de Infectología, Hospital Nacional de Niños “Dr. Carlos Sáenz Herrera”, Centro de Ciencias Médicas, Caja Costarricense de Seguro Social (CCSS), San José 10103, Costa Rica; 6Servicio de Gastroenterología y Nutrición, Hospital Nacional de Niños “Dr. Carlos Sáenz Herrera”, Centro de Ciencias Médicas, Caja Costarricense de Seguro Social (CCSS), San José 10103, Costa Rica

**Keywords:** rotavirus, children, acute gastroenteritis, seasonality, epidemiology, meteorology

## Abstract

Rotavirus is a leading cause of acute diarrhea in children worldwide. Costa Rica recently started universal rotavirus vaccinations for infants with a two-dose schedule in February 2019. We aimed to study the seasonality of rotavirus during the pre-vaccination era. We retrospectively studied a six-year period of hospital admissions due to rotavirus gastroenteritis. We estimated seasonal peak timing and relative intensities using trend-adjusted negative binomial regression models with the δ-method. We assessed the relationship between rotavirus cases and weather characteristics and estimated their effects for the current month, one-month prior and two months prior, by using Pearson correlation coefficients. A total of 798 cases were analyzed. Rotavirus cases predominated in the first five months of the year. On average, the peak of admissions occurred between late-February and early-March. During the seasonal peaks, the monthly count tended to increase 2.5–2.75 times above the seasonal nadir. We found the strongest negative association of monthly hospitalizations and joint percentiles of precipitation and minimal temperature at a lag of two months (R = −0.265, *p* = 0.027) and we detected correlations of −0.218, −0.223, and −0.226 (*p* < 0.05 for all three estimates) between monthly cases and the percentile of precipitation at lags 0, 1, and 2 months. In the warm tropical climate of Costa Rica, the increase in rotavirus hospitalizations coincided with dry and cold weather conditions with a two-month lag. The findings serve as the base for predictive modeling and estimation of the impact of a nation-wide vaccination campaign on pediatric rotaviral infection morbidity.

## 1. Introduction

Rotavirus is the most common cause of severe diarrhea among children younger than 5 years in both developing and developed countries [1]. The World Health Organization (WHO) estimates that rotavirus contributed 453,000 deaths within the global mortality of children <5 years in 2008 alone [2]. Prior to the introduction of rotavirus vaccine in 2006, approximately 15,000 deaths, 75,000 hospitalizations, and two-million medical visits were associated with rotavirus infection in Latin America and the Caribbean annually [3]. In Costa Rican children, rotavirus is the leading cause of acute viral diarrhea, accounting for 53% of rotavirus-positive stool samples predominantly with the G3 strain [4].

Rotavirus seasonality has been shown to differ widely across the world. In 1990, Cook et al. reported that rotavirus was common in colder months in temperate regions with a distinct winter peak in the Americas, different seasonal peaks in other temperate regions, and no distinct seasonality in the tropics [5]. A more recent meta-analysis shows that in the tropical belt, rotavirus cases have a lower seasonal intensity compared to more northern or southern latitudes, where prevalence peaks in winter months (November-April and May-October, respectively) [6]. In South Asia, the highest rate of rotavirus was seen in the colder, drier months [7]. In humid mid-latitude climates, low temperature and precipitation levels were significant predictors of an increased rate of rotaviral diarrhea. A 1 °C decrease in monthly ambient temperature and a decrease of 10 mm in precipitation were associated with 1.3% and 0.3% increase above the annual level in rotavirus infections, respectively. The relationships with temperature and precipitation in the previous month were strongest in the tropical climate [7].

Many authors have analyzed the association between rotavirus seasonality and weather parameters. Relationships between monthly rotavirus incidence and climatological variables, such as temperature, rainfall, and relative humidity, suggest that rotavirus infections tend to increase under cool and dry conditions in temperate [8] and tropical regions [7,9,10,11,12,13]. In contrast, rotavirus peaks often align with monsoon seasons in tropical settings [9]. This may be due to wider dispersal of the virus in floodwater, where contamination promotes infections with numerous enteric viruses, including rotavirus [14,15]. Kraay et al. demonstrated that in tropical climates, water sources play an important role in the environmental transmission of rotavirus at both the community and regional scales [16]. Environmental transmission risks may also be affected by temperature, river flow velocity, water sanitation, hygiene interventions, and water reservoir size [16]. It is likely that distinctly different seasonal patterns of rotavirus disease in locations with similar climate and level of development indicate a variety of factors affecting disease seasonality [6].

Since 2006, two rotavirus vaccines have been licensed in >100 countries worldwide. By the end of 2014, more than 70 countries had introduced the rotavirus vaccine into their routine childhood immunization programs [1]. Effectiveness of rotavirus vaccines against hospitalizations and severe diarrhea in Latin American children has been approximately 70%, with notable reductions in mortality rates in Brazil, Mexico, and Panama where the vaccination coverage was >80% [17]. Until recently, Costa Rica and Belize were the only two countries in Central America without universal rotavirus vaccination [18,19]. Costa Rica just started universal rotavirus vaccination with a two-dose schedule in February 2019. Age of first infection, strain distribution, the occurrence of mixed infections, seasonality, and risk of mortality can affect decisions about vaccine composition and delivery [20]. Understanding the seasonality of rotavirus in Costa Rica can help inform vaccination scheduling and resource management.

Our main objective was to describe the temporal pattern of monthly hospital admissions due to rotavirus gastroenteritis diarrhea during the six-year period from January 2010 to December 2015. We estimated peak timing and relative intensity of rotavirus hospitalizations and assessed the association between rotavirus prevalence in hospital settings and local meteorological parameters. A better understanding of rotavirus infection seasonality before vaccination will help to assess the efficacy of a nationwide vaccination campaign in the future. 

## 2. Data and Methods

### 2.1. Study Area

This study was conducted at the Hospital Nacional de Niños, Dr. Carlos Sáenz Herrera, the only national specialized pediatric tertiary referral and academic hospital of the Caja Costarricense de Seguro Social (CCSS), located in San José, Costa Rica. This city is both the nation’s capital and most populated province, receiving patients from all over the country, as it is part of the public health system. Its main population capture area is localized in the Central Valley, which includes the urban centers of San Jose, Heredia, Alajuela, and Cartago. Annual weather patterns are often described by the rainy and dry season due to Costa Rica’s location in the tropical belt 9.7489º north of the equator [21]. A mountain system divides the country into two hillsides, each with a different precipitation and temperature regimen following unique temporal distributions. The Central Valley is located on the west side of this mountain range, characterized by an annual mean temperature ranging from 19 to 22 °C, a relative humidity of 82%, precipitation ranging from 1500 mm to 2500 mm, and an average elevation of 1200 m above sea level. The region’s dry season consistently occurs between December and March, while the rainy season is between May and October (April and November serve as transition months between seasons) [22].

### 2.2. Health Outcomes

For this six-year retrospective analysis, the hospital admissions due to acute rotavirus gastroenteritis were retrieved from January 2010 to December 2015. Selection criteria included: (i) children less than 13 years of age, and (ii) laboratory-confirmed episode of acute rotavirus gastroenteritis among hospitalized patients. A physician evaluated all children with diarrhea who presented at the hospital. According to the child’s hydration state, he/she was either treated with oral rehydration as ambulatory patients or admitted and hydrated orally or intravenously in observation rooms. Those with severe dehydrating diarrhea or shock were transferred immediately to the emergency room. The last two categories (admitted to observation or emergency rooms) were considered as hospitalized patients. All nosocomial rotavirus infections were excluded from the analysis. Patients were identified retrospectively both from the microbiology laboratory database and by the Center’s Statistics and Medical Records Division. The patient’s clinical charts and laboratory reports were reviewed with respect to the inclusion criteria of the study. Rotavirus gastroenteritis was defined as three or more watery stools in 24 h, with or without vomiting and/or fever, with symptoms starting less than 15 days before admission. Patients also had to have a positive rapid antigen test performed within 48 h of admission. The VIKIA^®^Rota-Adeno immune-chromatography technique was used for diagnostic confirmation.

### 2.3. Meteorological Data

Weather data maintained by the National Meteorological Institute of Costa Rica was abstracted from monthly reports collected from 15 weather stations located in the Central Valley. Precipitation values were reported as monthly cumulative averages from thermo-pluvial-metric weather stations in the central region. Temperature values were reported as monthly averages of maximal and minimal temperatures from the same weather stations [23].

### 2.4. Statistical Analysis

All eligible hospitalization records were compiled into monthly time series of counts. We used two approaches to estimate seasonal peaks. First, seasonal peaks were defined as one or two consecutive months with the highest number of positive rotavirus tests in a given year. The onset of rotavirus season was defined as the first month when the number of monthly cases exceeded the overall median, and the duration of the season was defined as the number of months with positive results exceeding the median.

Next, to improve the estimation precision, we estimated seasonal peak timing, relative intensities and their 95% confidence intervals for all children, as well as younger (<24 months) and older (≥24 months) age-groups. We estimated peak timing and its variance using trend-adjusted Negative Binomial regression models with the δ-method [24,25], as follows:(1)$$\ln\left[E\left(Y_{\mathrm{tj}}\right)\right]=\beta_{0}+\beta_{t}t+{\;\beta }_{s}\left(\sin\left(2\pi \omega t\right)\right)+\beta_{c}\left(\cos\left(2\pi \omega t\right)\right)$$where Ytj –rotavirus cases for t-month if j-age group; t – refers to a time series indicating the consecutive study month from one to 72; sin(2πωt) and cos(2πωt) refer to two periodic terms defining seasonal oscillations with a frequency of ω = 1/12, representing the length of the annual cycle in months. The estimates of $\beta_{s}$ and $\beta_{c}$ regression coefficients and their error values were used to calculate seasonality characteristics: peak timing and relative intensity, e.g., the ratio between the predicted maximum and minimum counts.

In order to assess the relationship between monthly counts of laboratory-confirmed rotaviral infection and weather characteristics, we first examined the distribution of each meteorological variable. Next, we identified the suitable transformation and informative lags, or the time delay between the seasonal peak in rotavirus and the meteorological variables. Lastly, we examined the joint effect of temperature and precipitation on the health outcome. As expected, the distribution of monthly precipitation values appeared to be left-skewed. Thus, they were examined as percentile values in decimal continuous form ranging between 0 and 1, so the values are representing the whole range from the global minimum to global maximum, respectively. We then determined the cut point of 26.8 mm (the 25th percentile or 0.25 in decimal form) of precipitation to define the local dry period. Similarly, the local cold conditions were designated when the average minimum temperature dropped below 16.2 °C (the 25th percentile) (see Figure 1). We then identified the local dry and cold weather conditions jointly using the product of precipitation and mean minimum temperature percentiles (see Figure 1C).

Correlations were then estimated between the joint percentile for precipitation and minimum temperature for the current month, one month prior, and two months prior (or at lag 0-, 1-, and 2-) with monthly hospitalization counts using Pearson correlation coefficients. We repeated the calculations for children <24 months and children ≥24 months. These estimations allowed us to provide a robust and comprehensive assessment of the associations.

The data compilation and analysis were completed using Epidata 3.1 (The EpiData Association, Odense Denmark, 2004), Excel 2011 14.4.8 (Microsoft Corporation, Redmond, DC, USA), IBM SPSS Statistics version 21 (IBM Company, Armonk, NY, USA) and Stata 15 by (StataCorp, College Station, TX, USA).

The study was approved by the hospital’s bioethics and research committee.

## 3. Results

A total of 798 patients met the inclusion criteria during the six-year period. The mean age was 29.3 ± 26 months with 453 children aged <24 months and 345 aged ≥24 months. The age distribution included 63 cases for children 0 to <6 months (7.9%), 143 cases 6 to <12 months (17.9%), 373 cases 12 months to <36 months (46.7%), 112 cases 36 months to <5 years (14%), and 107 cases ≥5 years (13.4%). There were 478 male patients (59.9%). 

The average monthly values for rainfall and temperature recorded in the Central Valley during the study period are shown in Table 1. Overall, the average value for rainfall was 148.9 mm. The highest average monthly rainfall of 329.5 mm was observed in October, while January was the month with the lowest average rainfall of 12.5 mm. The average temperature was 21.2 °C, with the coolest month being January (20.4 °C), and the hottest months being May and July (21.9 °C). The greatest mean temperature gradient was observed on April (10.3 °C), when the mean maximal temperature was 27.1 °C and the mean minimum temperature was 16.7 °C.

Typically, rotavirus cases were highest in the first four months of the year, decreasing in June approximately two months after rainfall increases in April (Table 1 and Figure 2 and Figure 3). However, monthly counts of hospitalizations vary by year. The rotavirus season in 2010 started on October (with a sporadic spike in June) and extended through the first two months of 2011. The rest of the year 2011 presented with no marked season and the least amount of cases compared with the other years. In 2012, the rotavirus season started on April and extended to August, with the peak season on May, which also was the month with highest average precipitation in that year. This was an atypical year with respect to rotavirus incidence with most of the months having 6 or more cases. In 2013 and 2015, the rotavirus season lasted from January to April, and from January to May, respectively. The peak season in 2013 was in January and in 2015 was in March. Both peaks coincided with the months with the least precipitation. In 2014, the rotavirus season lasted from March to May with the peak of cases occurring in March. This variability is reflected by high values of the standard deviation for January and May (Table 1). 

Using negative binomial harmonic regression model, we estimated seasonality characteristics for all children and two age groups. The model results provide a robust estimation of peak timing and relative intensity, as shown in Table 2. Over the study period, hospitalizations due to rotavirus with an average peak between late-February and early-March for children in both age groups (all children: 2.96 (1.63, 4.30)). During the seasonal peaks, the counts are expected to increase 2.5–2.75 times higher than the seasonal nadir, calculated using relative intensity.

To further explore the relationship, we plot the monthly values of minimum temperature, precipitation, and rotavirus cases as parallel time series over the full study period (Figure 3). We highlighted the time periods of locally low temperature, or local cold conditions when the temperature dropped below 16.2 °C (the 25th percentile is shown as a continuous red line). We then overlaid the periods of locally dry conditions with the precipitation level below 26.8 mm (the 25th percentile is shown as a green line). This figure shows how these locally dry and cold periods are synchronized with elevated rotaviral cases defined by cases above the median level of 6 cases per month (the 50th percentile is shown as a light blue continuous line).

To examine how well the local dry and cold weather conditions coincided with monthly cases, we transformed meteorological variables to their percentiles and estimated their effects at 0-, 1-, and 2-month lags. We detected correlations of −0.218, −0.223, and −0.226 (*p* < 0.05 for all three estimates) between monthly cases and the percentile of precipitation at each lag, respectively. We then correlated the joint percentiles of precipitation and minimum temperature with monthly hospitalization counts at lag 0-, 1-, and 2- months. Across these three lags, the strongest negative association was observed at a lag of two months (R = −0.265, *p* = 0.027). We repeated the calculations for children <24 months and ≥24 months and confirmed these findings of R = −0.247, *p* = 0.039 and R = −0.264, *p* = 0.027, respectively (Figure 4).

## 4. Discussion

Although previous studies about rotavirus gastroenteritis among Costa Rican children have analyzed the monthly distribution of cases and peak timing during the year, we have provided a comprehensive analysis correlating rotavirus infection to three meteorological characteristics. As in previous local reports [26], we found a predominance of cases in the first half of the year, which locally coincides with dry and cold weather. For all but one year, cases increase between November and March, similar to the rotavirus season in the northern latitudes as reported by other authors [6]. On average, hospitalizations peaked between late February and early March, following or coinciding with the local dry weather in January, February, and March. Hieber and co-authors reported this phenomenon in the same study area more than 30 years ago [26]. Unusually, the rotavirus season of 2012 lasted from May to August and overlapped with the southern latitudes, coinciding with the rainiest months. 

The negative associations between rotavirus incidence and meteorological variables have already been reported in multiple studies run in both tropical and temperate areas [7,8,9,10,11,12,13]. In a similar tropical climate, in the city of Caricuao in Venezuela, a 6-year study in the pre-vaccine era demonstrated the year-round presence of rotaviral infection with a slight decrease during the wet and warm months [27]. A three-year surveillance study in Western Kenya with a predominantly tropical climate showed that stool samples collected during warm and dry months were twice as likely to be rotavirus-positive as compared to cool and rainy months, yet inconsistent across the years [28]. A meta-analysis of 26 studies conducted in the tropical climate between 1975 and 2003 demonstrated an inverse significant relationship between monthly rotavirus incidence and climatological variables such as temperature, rainfall, and relative humidity [6]. Unlike most of these published studies, we found a significant correlation at a two-month lag, indicating the delayed effects of cold and dry conditions on rotavirus incidence. Other authors have reported negative lagged correlations between rotavirus and both temperature and precipitation. Jagai and co-authors demonstrated that, in tropical and temperate regions of South Asia, rotavirus rates were the highest in the colder, drier months, based on negative significant relationships between rotavirus cases and temperature, precipitation, and vegetation index. The associations were prominent at no lag and one-month lag, yet strongest at one-month lag [7]. Hasan et al. recently analyzed rotavirus diarrheal risk due to hydroclimatic extremes and found that during the winter season, the rotavirus outbreak in Dhaka had similarly strong negative one-month lagged correlations with rainfall-related indices [29].

Numerous environmental, behavioral, and immunological mechanisms have been proposed to explain the seasonality of viral pathogens [30]. It is important to keep in mind that seasonal factors operate through many transmission mechanisms making it difficult to define an exclusive causal pathway. The issue of co-seasonality of numerous environmental, social, and behavioral phenomena may also influence the reproductive number of infectious diseases [30]. The influence of climatologic factors directly on rotavirus survival and infectivity is one example. Rotavirus survival is favored by cooler conditions with low humidity [6]. Given the characteristics of transmission of the virus (by fecal/oral route and by respiratory droplets), it has been hypothesized that a relative drop in humidity and rainfall combined with the drying of soils might increase the aerial transport of dried, contaminated fecal material [6]. At the same time, rotavirus can retain its infectivity for several days in aqueous environments and remain viable on inanimate surfaces when dried from fecal suspension [31]. Thus, these environmental phenomena may best be correlated with rotavirus infection using 1–2-month lag as proposed by some authors [7,29] and shown in our analysis. 

While many studies consider indirect transmission through water negligible when analyzing rotavirus transmission dynamics or were not able to accommodate this information, this form of environmental transmission may pose an important risk in tropical climates [16]. This could explain why rotavirus peaks weakly correlate with the rainiest months. Different weather variables can also influence human behavior, sanitation, hygiene practices and the probability of environmental exposures. For example, rainfall can encourage individuals to stay indoors, altering the pattern of contact between infected and susceptible individuals. 

In addition, research has suggested that annual epidemics or pandemics of different viral infections can interfere with each other, but clear trends over a long time and underlying mechanisms are unknown [32]. Although the additional mechanisms of transmission are beyond the scope of this paper, it is important to note that Costa Rica had a sanitation alert due to the first wave of H1N1 influenza (swine flu) on April 2009. In 2010, rotavirus peaked late on October, followed by very few cases in 2011, which was the year with the least number of cases. During this period, numerous sanitary measures, including prolonged school closures and the suspension of national gatherings, were undertaken and maintained. Also, there was great exposure in social media of preventive hygiene measures that should be followed to reduce the risk of acquiring the disease. Nevertheless, improvements in hygiene or sanitation have not proven to substantially reduce the burden of rotavirus disease [31]. 

This study did not analyze the genotypes of rotavirus strains, but strain distribution and variability can also be responsible for rotaviral outbreaks and changes in seasonal patterns. Previous investigations in our center had demonstrated that there was a great diversity of G and P genotypes circulating simultaneously with a frequent occurrence of unusual G-P combinations. Bourdett-Stanziola and co-authors analyzed rotavirus genotypes in Costa Rica from December 2002 and July 2003 and found that the most prevalent combination was G3P [8] (54% prevalence) [33]. A few years later, that same author analyzed rotavirus genotypes between August and October of 2005, and between May and June of 2006 and found G1P [8] and G9P [8] as the most prevalent combinations in each period, respectively [34]. 

Undoubtedly, vaccination against rotavirus is the best measure to prevent rotavirus disease. While rotavirus vaccine had not been introduced in Costa Rican universal immunization program at the time of the study, it has been available in the private practice. As this was not measured, immunization programs may have covered a minor fraction yet unknown percentage of our population. Our study provides the benchmark for seasonality of hospitalizations in the main pediatric center in the pre-vaccine period. We will compare these findings with the post-vaccination variations. We expect changes in seasonal patterns after universal vaccine implementation, including delays in the start of rotavirus season, shorter duration of seasons and blunting of seasonal peaks [35,36,37,38]. The presented model for estimating the peak timing will allow us to monitor changes in disease incidence.

Demographic structures also influence the seasonality pattern. We expect that by studying seasonal pattern in different age groups will help to describe changes in rotavirus incidence associated with vaccination. For example, in Finland, rotavirus infection was most common in children <5 years of age before vaccine introduction, but after the vaccine was introduced, the main age groups affected by rotavirus were children between 6–16 years of age and individuals >70 years of age [39]. Studies suggest that higher birth rates have been related to the lack of seasonality [40]. In a birth cohort study, the effect of age was found to be the most significant contributor for rotavirus incidence, showing a strong negative association, yet seasonality was well maintained [41]. 

Finally, seasonality in the tropics may also be driven by very complex weather conditions not measured in our study, like wind patterns [42]. Costa Rica is located under the influence of the inter-tropical convergence zone and equatorial winds [43]. The presence of natural phenomenon, like ENOS (El Niño Southern Oscillation), which is the most dominant source of inter-annual climatic variability in the Tropics [44], may influence the seasonality of different infectious agents. In June 2010, a strong La Niña pattern had started and was described as the most intense in the last 30 years. This pattern resulted in a mild increase in average precipitation in the Central Valley, lasting until February 2012. In April 2014, the region experienced a low-intensity El Niño pattern, which by May 2015 had intensified and lasted for the whole year [45]. The associations between diarrheal diseases and El Niño pattern were noted in other studies. For example, Checkley et al. reported a 200% increase in the daily number of expected admissions for diarrheal diseases of Peruvian children during the El Niño episode in 1997–1998 [46].

We acknowledge limitations of our study. First, because of its retrospective nature, we lacked important patients’ individual information, including exact home address, socioeconomic status, water supply, and overcrowding conditions. Second, we only examined the correlation between rotavirus and two weather variables. There is some variability in measuring precipitation across weather stations, however, it is unlikely that such variability affects our findings. The study site is located in the central region, one of the seven national climatic regions, and has a high density of thermo-pluvio-metric stations with the most reliable meteorological monitoring in the country. Furthermore, the detected peak of rotavirus hospitalizations occurred during dry months, when the amount of precipitation (and thus, its variability) is low. In locations with a limited ground meteorological network, satellite imagery could be used as the proxy [7,29]. Otherwise, we recognize a dual benefit of using data from local authorities. First, it signals to the weather forecast community that their information is of a high value to public health professionals. On the other hand, the public health community could be more proactive in working with local forecasters on delivering key warning messages in a timely manner.

We analyzed only children with rotavirus requiring hospital admissions and therefore the majority of less severe cases were not included. We may have also missed rotavirus-associated diarrhea hospitalizations, as we didn’t analyze all-cause diarrhea hospitalizations. Lastly, we examined cases from a single center, however, due to Costa Rica’s size and the hospital’s population catchment areas, these findings probably reflect similar conditions in other parts of the country. A better understanding of pathogen-specific infection seasonality is essential to target local and national health policies as well as design and implement well-tailored preventive strategies to control rotaviral infections especially when suitable vaccines are available.

## 5. Conclusions

At the main pediatric hospital in Costa Rica, there is a higher prevalence of rotavirus acute gastroenteritis hospitalizations in the first half of the year, with an average peak between late-February and early-March. During seasonal peaks, the counts are expected to increase 2.5–2.75 times higher than the seasonal nadir. In a warm tropical climate of Costa Rica, we observed significant inverse correlations between rotavirus admissions and cold and dry weather conditions with a two-month lag. With the routine assessment of temporal trends of potentially preventable infections and their seasonal uptakes, we will be able to better assess the effect of a broad implementation of vaccination and relevant prevention measures.

## Figures and Tables

**Figure 1 ijerph-16-02321-f001:**
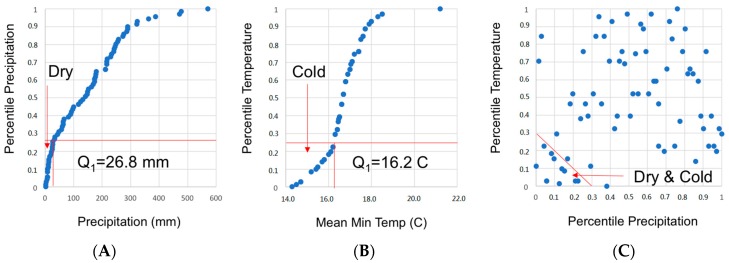
Scatter plots illustrating relationships between monthly precipitation values and their percentiles (**A**), monthly temperature values and their percentiles (**B**), and the joint percentiles for mean minimum temperature and precipitation identifying the cold and dry regiment (**C**), for 2010–2015 in the Central Valley of Costa Rica.

**Figure 2 ijerph-16-02321-f002:**
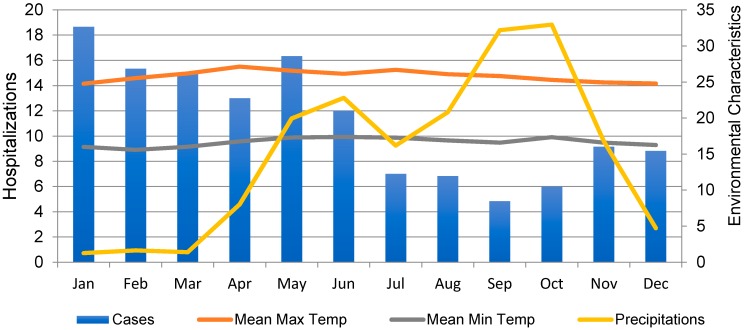
Average monthly counts for rotavirus hospital admissions and average environmental characteristics: rainfall (in cm), a mean maximal and minimal temperature (in °C) in 2010–2015 in the Central Valley of Costa Rica.

**Figure 3 ijerph-16-02321-f003:**
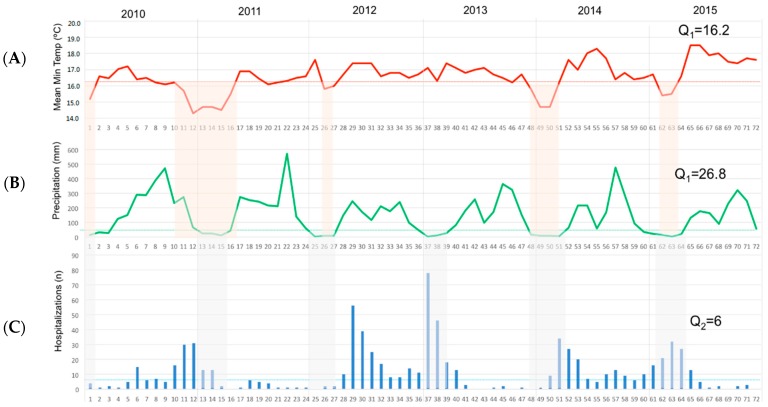
Time series on monthly values of minimum temperature (**A**), cumulative precipitation along with the periods below their 25th percentile values (**B**), and cases of hospital admissions (**C**) in 2010–2015 in the Central Valley of Costa Rica. Quartile 1 level for 25th percentile of minimal temperature (Q_1_ = 16.2 °C) is represented as a continuous red line; for 25th percentile of precipitation (Q_1_ = 26.8 mm) as a green line, and Quartile 2 level for 50th percentile of monthly cases (Q_2_ = 6) as a light blue line.

**Figure 4 ijerph-16-02321-f004:**
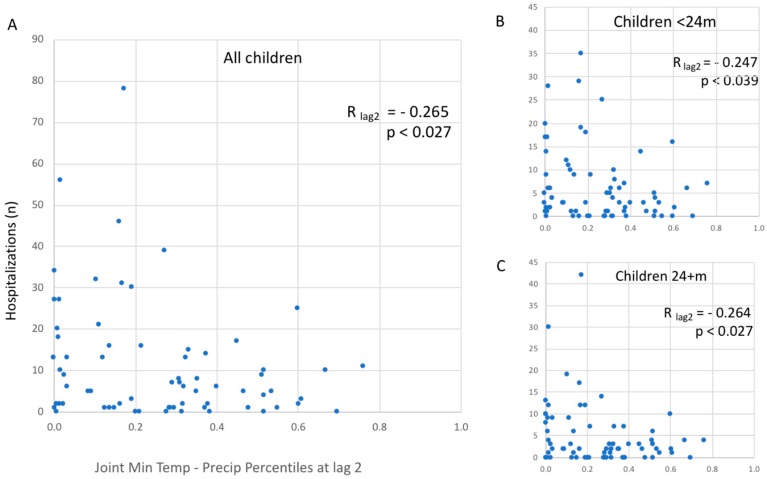
Scatter plots illustrating relationships between monthly rotavirus gastroenteritis hospital admissions and the joint percentiles for mean minimum temperature and precipitation at the lag of two months for all hospital admissions (**A**), for hospital admissions of children less than 24 months of age (**B**), and for hospital admissions for children 24 months and older (**C**).

**Table 1 ijerph-16-02321-t001:** Summary statistics for monthly values (mean and standard deviation) of rotavirus counts and meteorological parameters: precipitation, minimum and maximal temperature, for 2010–2015 in the Central Valley of Costa Rica.

Month	Rotavirus Counts		Precipitation (mm)		Minimum Temperature (°C)		Maximal Temperature (°C)	
	*Mean*	*SD*	*Mean*	*SD*	*Mean*	*SD*	*Mean*	*SD*
January	18.6	29.78	12.6	9.39	16.00	1.29	24.77	0.92
February	15.3	16.74	16.4	9.97	15.58	0.80	25.53	0.92
March	15.0	15.27	13.9	10.28	16.01	0.97	26.18	0.89
April	13.0	11.95	80.2	48.38	16.76	0.71	27.13	0.26
May	16.3	20.68	199.5	55.11	17.30	0.63	26.57	0.75
June	12.0	14.10	228.0	47.25	17.37	0.77	26.13	0.56
July	7.0	9.14	161.9	88.29	17.27	0.75	26.68	2.44
August	6.8	5.98	208.1	99.23	16.88	0.79	26.07	0.71
September	4.8	4.95	321.9	133.86	16.58	0.51	25.82	0.91
October	6.0	6.16	329.6	124.13	17.35	1.94	25.28	1.15
November	9.1	11.30	167.8	77.04	16.58	0.65	24.93	1.18
December	8.8	11.96	47.2	19.16	16.25	1.11	24.78	1.41
Overall	11.1	14.31	148.9	129.7	16.66	1.08	25.82	1.29

SD = standard deviation.

**Table 2 ijerph-16-02321-t002:** The peak timing and intensity values estimated from the regression model.

Study Group	Peak Time (month)			Relative Intensity (counts)		
	*Estimate*	*LCI*	*UCI*	*Estimate*	*LCI*	*UCI*
0–23 months	3.12	1.55	4.68	2.47	1.50	3.45
24+ months	2.89	1.65	4.13	2.68	1.51	3.85
All children	2.96	1.63	4.30	2.75	1.54	3.96

LCI = lower confidence interval; UCI = upper confidence interval.

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
