# Peer review of "Seasonality of Rotavirus Hospitalizations at Costa Rica’s National Children’s Hospital in 2010–2015"

_ijerph, 2019, doi:10.3390/ijerph16132321_

Round 1

Reviewer 1 Report

Presented manuscript describe seasonality of rotavirus in Costa Rica. A single center, retrospective study was conducted for a period of 2010 to 2015. Study represent disease seasonality before inclusion of rotavirus vaccination in Costa Rica schedule in Feb 2019. 

The study aims to define a method for estimation of peak of the disease and thus help health care centers to be ready. However, no such guidelines are provided clearly, rather with ambiguity. Further, analysis of data should have followed more stringent and widely accepted methods.

Major comments:

1.     Meteorological data was collected from different local weather stations. Variability of different weather stations should have been identified. With public availability of satellite-based data, such as Global Precipitation Measurement (GPM); such data set should have been utilized to detect any spatial variability and for validation of locally collected data.

2.     It is not clear if data in Table 2 is speculative (estimated) or calculated (from the present study)? If estimated, the result section (line 221-225) should reflect it and compare with the calculated observations. It will help in developing a model for disease prediction.

3.     Fig. 3, time series should have label such as Jan 10, Feb 10…Jan 11, Feb 11…etc. Representing month and year. Numbering them from 1-72 is not very helpful. Also, rather than minimal temperature, mean temperature for the month is a better indication of disease seasonality. It is not clear if precipitation shown is minimal or mean.

4.     In the last section of the results (line 248-255), results are presented as percentiles and an estimate of time lag had been calculated. The univariate correlation analysis of the result would have provided a better index, thus helpful in deriving conclusion. As done by Hasan et al in Ref. No. 27.

5.     Figure 4 does not aid in the understanding or interpretation of data presented. Therefore, could be removed.

6.     Since vaccine is given at child birth, how seasonality of infection will define vaccine implementation program and what preventive measures should be taken by authorities. These points should have been discussed.

7.     The discussion section should also compare seasonality of rotavirus in other countries. That would have given this study a global perspective.

8.     In conclusion section, aim of the study is not reflecting.

Minor comments:

1.     Fig.1. Figure legend should mention that data points represent 6 years of observation, as done for Table 1.

2.     Similarly, In result section (line 185-188), it is not clear if case numbers are for 6 years period.

3.     Manuscript should be proofread for typos. E.g. Line 69…”contaminated” should be “contamination…”.

Author Response

Reviewer 1.

Major comments:

1.      Meteorological data was collected from different local weather stations. Variability of different weather stations should have been identified. With public availability of satellite-based data, such as Global Precipitation Measurement (GPM); such data set should have been utilized to detect any spatial variability and for validation of locally collected data

We are thankful for the comment. With respect to variability of different weather station it is important to note that the study site is located in the central region of Costa Rica, one of the 7 national climatic regions. This region has the highest density of thermopluviometric stations and the highest monitoring stability of the meteorological records in the country (personal communication with Dr.Wegner Scholz, Director of Weather ForecastUnit in the Instituto Meteorológico Nacional de Costa Rica).  While the variability across the station is likely, especially for precipitation data, so the variability in population density and locations from where patients can come from is also likely. Clearly, not the rain itself make people sick, but the consequences of floods and droughts, affecting the exposure routes. We found the rotavirus hospitalizations peaks during dry months, when the amount of precipitation (and thus, its variability) is low. The effect of high variability in precipitation measurements could be of high concern for infections that coincided with wet seasons, like cholera.  The intrinsic failure to capture spatial heterogeneity is typical for all studies of such nature, when the measure of a specific exposure to a pathogen is not feasible and various proxies are used instead. We commented on this issue in Discussion (lines509-515). 

We agree with the Reviewer that with the advent of remote sensing (RS), information on meteorological data become available to researchers (Jagai et al and Hasan et al). GPM data are especially valuable in the areas where local ground measurements are nor readily available. However, in the locations where weather records are plentiful and reliable, the use of data from local authorities could provide a dual benefit. It signals to weather forecast community that information they collect is of high value to public health. On the other hand, the public health community will be more proactive in drafting key messages and work with local forecasted on delivering proper warning.  We commented on this issue in Discussion (lines515-518). 

We see the value in further calibrating ground measurement and satellite imagery, especially in the areas of complex topology. In a validation study of one of the satellite estimation methods (CHIRPS v.2) in Costa Rica, a good correlation between ground and satellite-based estimates was detected for the station located in the central climatic region in the metropolitan area, but not for the other areas of the country (Rojas N. Validation of the CHIRPS rainfall database v.2 for Costa Rica on a monthly scale for the period 1981-2013– provide journal, year, or mention personal communication). While the validation is beyond the scope of presented study, we recognize the potential of RS data. Although, the estimates of rainfall generated from satellite images do not detect precipitation as such, they are valuable proxies where rainfall information is lacking (Tote C, Patricio D, Boogaard H et al., Evaluation of satellite rainfall estimates for drought and flood monitoring in Mozambique, Remote Sensing, 2015; 72: 1758-1776).  Estimates of precipitation generated by satellite images are still subject to potential algorithmic biases and uncertainties (associated with the superior reflectance of the cloud, thermal radiation, frequency of transfer of the satellite and the terrain geography), due to the indirect nature of satellite measurements. Nevertheless, we commented onusing satelliteimagery when the ground network is insufficientin Discussion (lines514-515). 

2.      It is not clear if data in Table 2 is speculative (estimated) or calculated (from the present study)? If estimated, the result section (line 221-225) should reflect it and compare with the calculated observations. It will help in developing a model for disease prediction.

We agree with Reviewer that the model used for estimating disease peak timing could be valuable for predictive purposes in reflected this in Discussion (line485-486). The statement related to distinction between “speculative (estimated) or calculated (from the present study)” is puzzling. We had estimated the peak time values along with the confidence intervals using the observed monthly counts of hospitalization. The model is described in Methods (lines165-169). We provided a clarifying sentence at the beginning of the paragraph describing the results in Table 2 (lines268-270).   

3.      Fig. 3, time series should have label such as Jan 10, Feb 10…Jan 11, Feb 11…etc. Representing month and year. Numbering them from 1-72 is not very helpful. Also, rather than minimal temperature, mean temperature for the month is a better indication of disease seasonality. It is not clear if precipitation shown is minimal or mean.

We cordially disagree with Reviewer that numbering 1-72 is not helpful. It serves the purpose to remind that all records were used in the model for peak timing estimation. Since we had provided the separators for individual years in Figure 3 and offered Figure 2 to illustrate a common seasonal pattern, the indication of month is unnecessary and makes labeling horizontal axes excessive. Precipitation values were reported as monthly cumulative averages (lines 139-140); we clarified this in theFigure3legend (line 296).

4.      In the last section of the results (line 248-255), results are presented as percentiles and an estimate of time lag had been calculated. The univariate correlation analysis of the result would have provided a better index, thus helpful in deriving conclusion. As done by Hasan et al in Ref. No. 27.

We appreciate Reviewer suggestions; however, we refrain from changing the approach because the substantial differences in study design and study objectives in our and Hasan et al papers. Hasan et al study was conduced in a much larger setting of Bangladesh, aiming to select meteorological indexes suitable for large geographic areas and detect differences in vulnerability of dry-cold regions of the country, compared to the wet-warm regions. Dhaka’s population alone is 8M, while San Jose is barely above 300,000 residents. In the context of the presented study, the value of examining 34 climatic indexes is limited. Secondly, we demonstrated the values of operating with the joint characteristic of air temperature and air water content, such as joint percentile values as describes and clarified in Figure 4.   To clarify, we are not presenting the results as percentiles, but rather using joint percentiles to demonstrate the relationship between disease incidence and weather characteristics.

5.      Figure 4 does not aid in the understanding or interpretation of data presented. Therefore, could be removed.

We recognize the educational value of any research publication and believe that the concept of the joint percentile values could be novel for epidemiologists and public health professionals, the main audience we are targeting.  as provided justification for keeping Figure 4 as described above.

6.      Since vaccine is given at child birth, how seasonality of infection will define vaccine implementation program and what preventive measures should be taken by authorities. These points should have been discussed.

Wedon’t expect that by knowing seasonal pattern thepractice of vaccination change. Weratherexpect thatthedemographic structure of patients with rotavirus might be altered. We also expect thatpublic health comminution could be better tailoredand delivered before seasonal outbreaks. Weexpanded the Discussion,pleasesee lines431to485and 487 to 491.

7.      The discussion section should also compare seasonality of rotavirus in other countries. That would have given this study a global perspective.

We expanded this topic in the Discussion, please see lines339-347.

8.      In conclusion section, aim of the study is not reflecting.

We hadaddthe text in the conclusionto better describe our objective(lines 529-649).

Minor comments:

1.      Fig.1. Figure legend should mention that data points represent 6 years of observation, as done for Table 1.

Corrected.

2.      Similarly, In result section (line 185-188), it is not clear if case numbers are for 6 years period.

Corrected.

3.      Manuscript should be proofread for typos. E.g. Line 69…”contaminated” should be “contamination”

Thank you for noting. Corrected.

Author Response

Reviewer 2.

1. The number " 0.223" should be " -0.223" on Abstract.

Thank you for noting. Corrected.

2. The "9.7489oNorth of the equator" should be modified as "9.7489oC North of the equator" on line 101, page 3.

We use international standards to provide the GPS geocoded location of the city, which is: 9.7489º N, where N stands for North of the equator. GPS coordinates are a unique identifier of a precise geographic location on the earth, usually expressed in alphanumeric characters with the prime meridian is 0 degrees longitude and the locations are measured according to 90 degrees east or west from that point.

3.      The "characterized by an annual mean temperature of 19.22oC "should be "characterized by an annual mean temperature of 19.22oC " on line 104, page 3.

We clarified that we meant the range of temperatures and modified the text. The symbol is corrected.

4.      The "when average minimal temperature dropped below 16.2oC " should be modified as"when average minimal temperature dropped below 16.2oC " on line 163, page 4.

The symbol is corrected.

5.      The "Q 116.2C "should be modified as " 1 16.2o Q  C" on Figure1B, page 4 and line 244, page7. In Figure 3, “ 1 Q 16.2C” is wrong too.

The symbols are corrected.

6.      Lines 251-253, page 7, the sentence "We then correlated the joint percentiles of precipitation and minimal temperature with monthly hospitalization counts and found the strongest negative association at a lag of two months, (R=-0.265, p=0.027)." has " the strongest negative association". What is the superlative in the sentence compared with ?

We corrected and clarified the sentence.

7. The formation should be improved. There lack some punctuation in this paper. For example, the period should exist after "shown in our analysis" on line 298, page 9, and so on.

Corrected.

8. Please check the form and correctness of the references carefully.

For example, “Romero, C.; Mamani, N.; Halvorsen, K.; Iñiguez. V. Enfermedades diarreicas agudas asociadas a rotavirus. 398 Rev Soc Bol Ped. 2005, 44, 75-S2.” is wrong.

Corrected.